

# Variability in deer diet and plant vulnerability to browsing among forests with different establishment years of sika deer

Yuzu Sakata, Nami Shirahama, Ayaka Uechi and Kunihiro Okano

Biological Environment, Akita Prefectural University, Akita, Japan

## ABSTRACT

Increased ungulate browsing alters the composition of plant communities and modifies forest ecosystems worldwide. Ungulates alter their diet following changes in availability of plant species; however, we know little about how browse selection and plant community composition change with different stages of deer establishment. Here, we provide insight into this area of study by combining multiple approaches: comparison of the understory plant community, analysis of records of browsing damage, and DNA barcoding of sika deer feces at 22 sites in forests in northern Japan varying in when deer were first established. The coverage of vegetation and number of plant species were only lower at sites where deer were present for more than 20 years, while the difference in plant coverage among deer establishment years varied among plant species. Deer diet differed across establishment years, but was more affected by the site, thereby indicating that food selection by deer could change over several years after deer establishment. Plant life form and plant architecture explained the difference in plant coverage across establishment years, but large variability was observed in deer diet within the two categories. Integrating these results, we categorized 98 plant taxa into six groups that differed in vulnerability to deer browsing (degree of damage and coverage). The different responses to browsing among plant species inferred from this study could be a first step in predicting the short- and long-term responses of forest plant communities to deer browsing.

## INTRODUCTION

A recent global conservation issue is the drastic increase in ungulate populations that has caused degradation in forest ecosystems. For example, high densities of red deer (*Cervus elaphus*) in Europe and New Zealand, moose (*Alces alces*) and white-tailed deer (*Odocoileus virginianus*) in North America, and sika deer (*C. nippon*) in Japan are caused by recent introductions or environmental changes (*Côté et al., 2004*; *Cooms et al., 2003*; *McCullough, Takatsuki & Kaji, 2009*; *Tape et al., 2016*; *Kaji & Iijima, 2017*). Increased deer browsing alters the composition of plant communities and decreases the diversity of plant species within one or two decades in areas such as North America, and

Corresponding author
Yuzu Sakata, sakatayuzu@gmail.com

the Netherlands (*Akashi & Nakashizuka, 1999*; *Russell, Zippin & Fowler, 2001*; *Maesako & Takatsuki, 2015*; *Ramirez et al., 2019*). However, temporal changes in the plant community in response to browsing have not been fully assessed as most studies are based on a comparison of the vegetation before and after ungulate population control or exclusion by fencing. These studies have shown that the recovery of vegetation often takes decades after a decrease in deer density, or the degradation of forest vegetation may be irreversible (*Mysterud, 2006*; *Tanentzap, Kirby & Goldberg, 2012*; *Tamura, 2016*).

Deer populations are usually managed using estimated densities and perceived carrying capacity (*Kaji et al., 2010*). The estimation of deer densities is technically demanding requiring extensive field sampling or elaborate population models (*Millspaugh et al., 2009*), while recent studies show improvement and new developments in population estimation (*Iijima, 2020*). Moreover, deer impacts can be low or high at the same density, depending on forage availability and browsing history, which is critical to the regeneration of plant species (*Tanentzap, Kirby & Goldberg, 2012*). Recent studies have emphasized the need to monitor not only deer density but also its impact on the ecosystem, including vegetation (*Stout et al., 2013*; *Iijima & Nagaike, 2015*). Plant responses to grazing are known to differ among species which may be explained by plant architecture and traits, due to food selection by deer, tolerance to herbivory, and competitive interactions among plants (*Augustine & DeCalesta, 2003*; *Hanley et al., 2007*; *Diaz et al., 2007*; *Nishizawa et al., 2016*). As plant persistence under browsing pressure may vary among plant species, both food selection by deer and plant abundance must be considered to understand plant vulnerability to deer browsing.

It has been shown that deer change their diet following changes in vegetation availability over several decades or even short periods (*Brown & Doucet, 1991*; *Bruinderink & Hazebroek, 1995*; *Takahashi & Kaji, 2001*). They can also change their consumption based on food availability at the patch scale including neighboring plants (*Bergval et al., 2006*; *Rautio et al., 2012*). Investigation of the diet of ungulates in addition to observed browsing damage would be effective in revealing deer-food selection because browsing damage may include damage caused by untargeted ungulate species. Conventional methods of diet analysis, including analysis of gut contents or microscopic analysis of feces, require considerable training and extensive reference plant collection (*Takatsuki, 1978*; *Takatsuki, Fuse & Ito, 2010*), and the absence of a unique epidermal structure in plant species leads to low taxonomic resolution (*Pompanon et al., 2012*). Conversely, DNA barcoding with feces is reported to have a high taxonomic resolution and is a useful and fast technique to estimate herbivore diets including those of ungulates (*Valentini, Pompanon & Taberlet, 2009*; *Raye et al., 2011*; *Ando et al., 2013*; *Kress et al., 2015*; *Nakahama et al., 2020*). However, whether it can be used for quantitative evaluation is still under consideration (*Pompanon et al., 2012*). It is important to combine multiple approaches for a robust evaluation of ungulate diet.

In forests in Japan, the population of sika deer, (hereafter referred to as deer) has drastically increased in the last two decades, and its distribution range has expanded by more than 2.5 times in the past 36 years from 1978 to 2015 (Fig. 1A, *Ministry of the Environment, 2015*). Especially in the northern Tohoku region of Japan, deer were almost

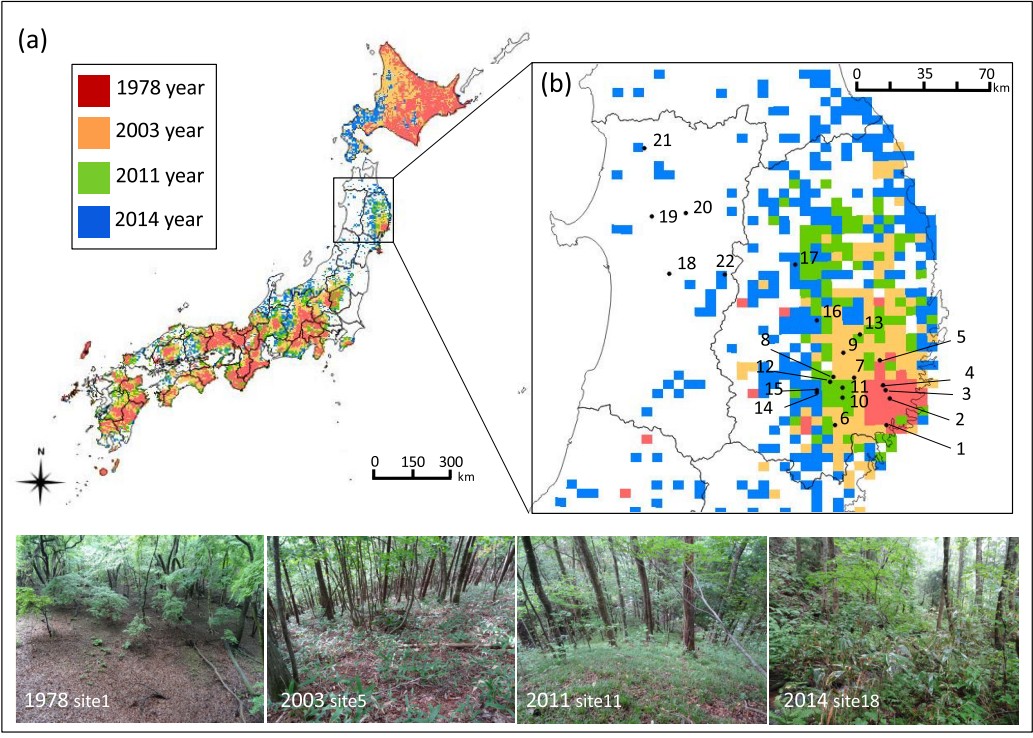

**Figure 1 Geographic location of study sites in northern Japan and the five km meshed area categorized into four sika deer establishment years.** Geographic location of study sites in Japan and (B) in the northern Tohoku region indicated by numbers and the five km meshed area categorized into four establishment years of sika deer according to the Japanese Ministry of the Environment: (1) deer established before 1978 (1978 year site), (2) deer established during 1979–2003 (2003 year site), (3) deer established during 2004–2011 (2011 year site), (4) deer established during 2012–2014 (2014 year site), and (5) deer not established. The site numbers correspond to the description in Table S1. Areas with no mesh indicate areas with no records of deer establishment. Although sites 18–22 are located in the unestablished meshed area, deer were observed at all sites, and thus, these sites were categorized as (4). Photos show the typical forest floor of each establishment year.

extirpated in the early 1900's but the population has increased rapidly since 1950 (*Takatsuki, 2009b*), and its distribution range has expanded, especially during the 2010's (*Ministry of the Environment, 2015*). With the growing demand to address the serious threats posed by sika deer, techniques for population monitoring that can be applied to low deer densities are required during the initial stage of establishment in these areas. Recent studies have introduced a novel approach to evaluate the abundance of sika deer in low-density populations, such as methods to distinguish between feces of sika deer and Japanese serow (*Aikawa, Horino & Ichihara, 2015*), and the acoustic monitoring of male abundance (*Enari et al., 2017*). However, there is a lack of knowledge about the response of plant communities in the initial stage of deer establishment. Information on the response of plant species to browsing after the initial stage of deer establishment would be informative for conservation efforts to maintain plant community diversity. The years since deer establishment exhibits geographic variability due to the recent expansion of deer in the Tohoku region. This provides a promising opportunity to

compare food selection among different establishment years of deer, and examine plant vulnerability to browsing from the initial stage of deer establishment. We investigated the understory plant community and deer diet at 22 sites in forests in northern Japan, that vary in year of deer establishment. We predicted that deer diet and plant community composition would vary among deer establishment years. We evaluated plant vulnerability based on both deer-food selection (degree of damage) and vegetation (plant cover) among plant species in response to deer establishment.

## MATERIALS & METHODS

### Study sites

This study was conducted from 2016 to 2017 at 22 cool-temperate deciduous-forest sites at an average elevation of 410 m dominated by oaks such as *Quercus crispula*, and *Q. serrata* in the northern Tohoku region in Japan (Table S1, Fig. 1B). We focused on this type of vegetation distributed in low-elevation forests because this allowed us to replicate multiple sites across the region. According to the Japanese Ministry of the Environment, Japan is divided into five km meshed units, which are categorized into five categories: (1) deer established before 1978 (1978 year site), (2) deer established during 1979–2003 (2003 year site), (3) deer established during 2004–2011 (2011 year site), (4) deer established during 2012–2014 (2014 year site), and (5) not established. We replicated 4–9 sites for each category of (1) to (4). The estimated deer density of the five km meshed unit that contained each site was higher in older deer establishment sites, while it varied in recent establishment sites (Table S1). Although deer density may vary between years, we selected sites where deer have been constantly observed in the two years before the survey according to the *Tohoku Regional Forest Office (2017*; approval number: 29-392, 29-487, 28-446*)*. Although sites 18–22 are located in meshed units categorized as (5), deer were observed in these sites in the 2 years before the survey, and thus, these sites were categorized as (4). We did not include sites in category (5) because our main objective was to compare the food selection of deer among establishment years and we cannot evaluate this in sites without deer.

### Vegetation survey of forest understory

To investigate the plant community of the site (500 × 500 m), we set 20 random replicates of 2 × 2 m plots in the forest floor including both ridges and valleys at each site in early September. Surveys were conducted in either 2016 or 2017. Plants with more than 5% coverage were recorded at 5% intervals, and those with less than 5% coverage were scored as 1% following the method used by *Sakata & Yamasaki (2015)*. We included flowering plants that were <130 cm in height. We excluded ferns because little browsing damage was found on ferns. However, it is notable that some unpalatable ferns can become dominant in intensively browsed forests (*Cretaz & Kelty, 1999*; *Ishida et al., 2008*), which may affect the plant community. We categorized each plant species into each life form (annual, perennial, vine, tree, shrub). We also categorized forb species by plant architecture (prostrate, erect, rosette, rosette [seasonal], tussock) according to the indices

described by *Asano (2005)*. Note that the categorization of woody species of plant architecture is identical with the categorization of life form.

## Browsing damage investigations

To investigate browsing damage of deer across plant species, we measured damage to focal plant species at each of the 22 sites in early September. We first assessed the characteristics of the browsing damage to exclude damage by other herbivores such as rabbits (rabbits leave clean cuts, while deer leave rough or torn cuts). Although we cannot completely exclude the possibility that the browsing damage was made by Japanese serow (*Capricornis crispus*), we only found feces of deer in our sites, as described in the feces sampling section below. Therefore, the browsing damage found in our survey was likely caused by deer. We selected 273 species (86 families) as focal plant species to cover the plant species that appeared in the vegetation survey of all sites. We randomly selected 10 individuals in the forest understory for each plant species which were present by walking around the whole area of each site including the 20 plots of the vegetation survey. Each plant individual was separated in more than three m. We aimed to evaluate browsing damage in the growing season of the plant, which is during spring and summer of the current year. Each plant species was categorized by life form and plant architecture. For each plant unit (ramet for forbs and current-year shoots for woody plants and vines), the damage level was visually estimated and classified into the following four levels: CL1, no damage; CL2, 1–10% damage of the plant; CL3, 11–50% damage of the plant; CL4, 51–100% damage of the plant. Subsequently, the values for all four levels were added and divided by 10 (*i.e.*, total number of plants or ramets: $N$) to calculate the mean damage grade (MDG) (*Sakaguchi et al., 2012*) for each plant species at each site, as shown in the following equation:

$$MDG = (0 * CL1 + 5 * CL2 + 30 * CL3 + 75 * CL4)/N$$

We estimated the damage level by assuming that the plant unit was in the averaged size of the undamaged plants observed at each site. Although this method may lead to biased evaluation among plant life forms, the broad classification (*i.e.*, four categories) would reflect the relative damage, which is comparable across plant taxa.

## Sampling feces and diet analysis using DNA barcoding

We prepared a trnL P6 loop region reference database comprising 297 flowering plant species that occurred at our study sites (Table S2); 167 species obtained from a database in a previous study (*Nakahama et al., 2020*), 71 species from the National Center for Biotechnology Information (NCBI), and 59 species were newly sequenced in this study (see Appendix 1 for detailed methods).

To evaluate the diet of deer, we searched for deer feces at each site and when available, fecal pellets were sampled from 5–10 dung piles (10 pellets on average per dung pile) found in each site during late July to early September and stored at −20 °C. We collected feces from 16 sites (Table S1). Note that for site Tazawa, feces were collected in November. Of these, we chose five fresh fecal pellets from 3–5 dung piles collected at each site.

Each five pellets correspond to one fecal sample, yielding 63 fecal samples in total. These pellets were checked whether they were deer feces using a recently developed loop-mediated isothermal amplification method targeting the cytochrome b gene in the mitochondrial DNA of deer (*Aikawa, Horino & Ichihara, 2015*). The food plant DNA was extracted from each fecal sample, obtained the sequences of the plant DNA, and identified the plant taxon in each fecal sample using the reference database described above (see Appendix 2 for detailed methods).

## Data analyses

We conducted multiple analyses of data obtained from each survey, as summarized in Table S3. All the generalized linear mixed model (GLMM) analyses were conducted using the lme4 package (*Bates et al., 2015*) and car package (*Fox & Weisberg, 2019*) of R 3.3.2 (*R Development Core Team, 2016*). In all the GLMM analyses described below, the significance of the variables was determined using a likelihood ratio test, compared to the chi-square distribution. When the effect of a variable was significant ($P < 0.05$), we tested for differences among categories within a variable using Tukey's honestly significant difference (HSD) test on least-squared means with the overall type I error rate at 5% using the "lsmeans" package (*Lenth, 2015*) of R. All the permutational multivariate analysis of variance (PERMANOVA) analyses and non-metric multidimensional scaling (NMDS) analyses were performed based on Bray–Curtis dissimilarity index. In all the PERMANOVA analyses we described, we also tested whether the multivariate dispersion regarding the group centroid differed among the main effects by using the betadisperser function in VEGAN package in R (*Oksanen et al., 2019*), and it was revealed that composition differed among the main effect. In all K-means clustering analyses, the number of groups was determined using gap statistics (*Tibshirani, Walther & Hastie, 2001*).

### Vegetation survey

We conducted the following six vegetation analyses. First, to evaluate how plant species diversity differed among deer establishment years, generalized linear models (GLMs) were applied. We set the number of species and the Shannon–Wiener diversity index per site as response variable, establishment year as an explanatory variable with Poisson and normal distributions, respectively. Second, to test whether the composition of understory species differed among deer establishment years, we performed a PERMANOVA. We used the data matrix of the 273 plant species coverage of each site (the coverage of the 20 plots were added), and we included deer establishment years of each site as an explanatory variable. For this analysis, the ADONIS function in the library VEGAN in R was used. We used NMDS to examine understory species composition. To consider spatial structures, we also used distance-based Moran's Eigenvector Maps (dbMEM) on detrended data and x,y coordinates using the adespatial R package (*Dray et al., 2021*). Significance of the spatial vectors was assessed using ANOVA and forward selection was carried out to identify significant dbMEM vectors following *Borcard, Gillet & Legendre (2018)*. Third, to investigate how vegetation cover differed among deer

establishment years, we used GLMMs with Poisson distribution. The sum of vegetation coverage of all plants per plot was set as response variable and deer establishment year as an explanatory variable. Site nested within surveyed year was included in the model as random intercept to deal with unknown effects of climate and resource variations among sites and surveyed years of the study. Fourth, to test whether the plant coverage differed among deer establishment years for each life form and plant architecture (excluding woody species), GLMMs with Poisson distribution were applied. Coverage of each life form and plant architecture per plot was set as the response variable and deer establishment year as an explanatory variable, and site nested within surveyed year was included in the model as random intercept. Fifth, to explore whether the patterns of variation in coverage among deer establishment years differed among plant life forms and plant architectures, a PERMANOVA was performed. For 58 plant species that were present in at least one site of each establishment year in the vegetation survey, coverage of each plant species for each establishment year was averaged over sites, yielding four coverage values per species. Subsequently, the effects of life form on the coverage of the four combined coverage values were tested. For the 30 species excluding woody species, the effect of plant architecture on the coverage of the four combined establishment years was tested. The coverage for each establishment year was averaged over the sites. Finally, a K-means cluster analysis of the coverage of the four establishment years combined for 58 plant species was performed to identify groups of plants with similar patterns of variation in the coverage among deer establishment years. The K-means clustering was set to define two statistically distinguishable groups.

### Browsing damage of plants

We conducted the following three analyses for deer browsing damage. First, to test whether damage level was affected by plant coverage, GLMM with normal distribution was applied. The squared transformed MDG (mean damage grade) of each species was set as a response variable and plant coverage as an explanatory variable. Site nested within surveyed year was included in the model as random intercept. Second, to explore whether patterns of variation in the MDG among deer establishment years differed among plant life forms and plant architectures, a PERMANOVA was performed. For 34 plant species that were present in at least one site of each establishment year, the effects of life form and plant architecture (excluding woody species) on the MDG of the four combined establishment years was tested. MDG of each plant species for each establishment year was averaged over sites, yielding four MDG values per species. Finally, a K-means cluster analysis of the MDG of the 34 plant species was performed to identify groups of plants with similar patterns of variation in the damage level among deer establishment years. The K-means clustering was set to define three statistically distinguishable groups. Additionally, we listed 16 species that were absent in the 1978 year sites but were browsed in sites within the other three establishment years because these plants may have disappeared in the 1978 year sites due to browsing.

### DNA barcoding of feces

We conducted the following three analyses for the DNA barcoding analysis. First, we performed a PERMANOVA to test whether diet composition differed among deer establishment years. We used the data matrix of the proportion of reads (*i.e.*, number of sequence reads of each plant taxa/the total number of sequence reads) of 98 plant taxa contained per sample. The effect of site was considered as a block effect by including site as a strata function in the library VEGAN in R. To consider spatial structures, we also used dbMEM on detrended data and x,y coordinates. Significance of the spatial vectors was assessed using ANOVA and forward selection was carried out to identify significant dbMEM vectors. In temperate deciduous forests, seasons of the year influences plant development stage and consequently food availability to deer (*Takatsuki, 1986*; *Dumont et al., 2005*; *Nakahama et al., 2020*); therefore, we also conducted the same analysis described above by excluding samples collected in Tazawa site in November. An NMDS was performed using the data matrix of the proportion of reads of each plant taxa to visualize the diet composition distances among fecal samples. Second, to test whether the proportion of reads differed among deer establishment years for each life form and plant architecture, GLMMs with normal distribution were applied. The proportion of reads of each plant category was set as the response variable and deer establishment year as explanatory variable, and site nested with in surveyed year was included in the model as random intercept. Third, to explore whether patterns of variation in the proportion of reads of each plant taxa among deer establishment years differed among plant life forms and plant architectures, a PERMANOVA was performed. The 98 plant taxa detected in the analyses were categorized by life form and plant architecture. We analyzed the effects of life form and plant architecture (excluding the woody species) on the proportion of reads of the four establishment years combined for the 98 plant taxa. The proportion of reads for each establishment year was averaged over sites. Finally, a K-means cluster analysis of the number of reads of the 98 plant taxa was performed to identify groups of plants with similar patterns of variation in the damage level among deer establishment years. The K-means clustering was set to define three statistically distinguishable groups.

## RESULTS

### Vegetation coverage of understory plant community

We detected 273 species (86 families) during the vegetation survey at 22 sites. In the 1978-year sites, deer browsing damage and feces were abundant, and the understory vegetation was scarce (Fig. 1). The number of plant species significantly differed among deer establishment years ($\chi^2$ = 40.99, $P < 0.001$, N = 22, d.f. = 3), and it was lower in the 1978-year sites than other sites (Fig. 2A). Conversely, the Shannon–Wiener diversity index did not differ among the establishment years (Fig. 2B, $\chi^2$ = 7.75, $P = 0.05$, N = 22, d. f. = 3). The sum of vegetation cover significantly differed among deer establishment years ($\chi^2$ = 48.55, $P < 0.001$, N = 440, d.f. = 3), and the 1978-year sites were lower than the other sites, and the 2014-year sites were higher than the other sites (Fig. 2C). Plant coverage differed among deer establishment years within life forms and plant architecture

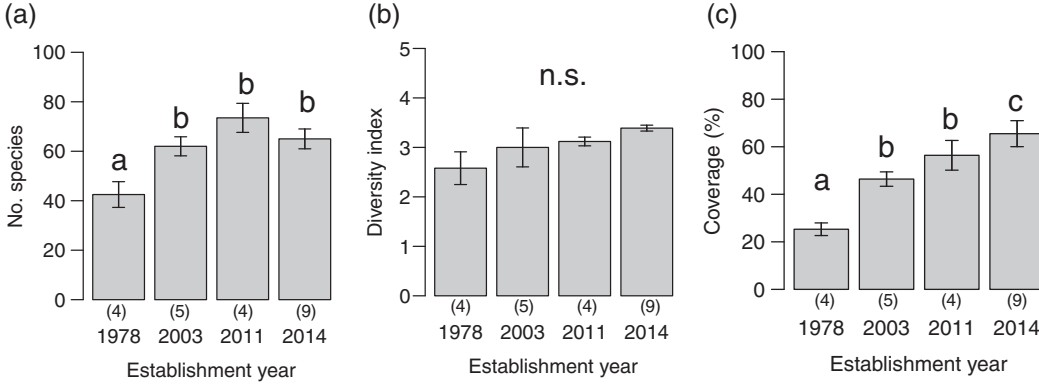

**Figure 2 Plant community indices (mean ± SE) of the four establishment years of sika deer.**
(A) Number of plant species per site, (B) Shannon–Wiener diversity index per site, (C) sum of vegetation coverage per plot. Different letters indicate significant pairwise difference after post-hoc adjustment among establishment years. n.s. indicate no significance difference ($P > 0.05$). The number of sites per establishment years are provided by the number in parenthesis above each year. For coverage, 20 plots were included per site.                                 

**Table 1 Results of the GLMM explaining the effect of deer establishment years on plant coverage of each categories.**

|  |  | Establishment year (d.f. = 3) | |
| --- | --- | --- | --- |
|  |  | $\chi^2$ | $P$ |
| Life form | Annual/biannual | 0.96 | 0.81 |
|  | **Perennial** | **14.01** | **0.003** |
|  | **Vine** | **20.08** | **<0.001** |
|  | Tree | 5.60 | 0.13 |
|  | **Shrub** | **28.21** | **<0.001** |
| Plant architecture | **Prostrate** | **10.32** | **0.016** |
|  | Errect | 3.56 | 0.31 |
|  | Rosset | 2.68 | 0.44 |
|  | Rosset (seasonal) | 6.15 | 0.10 |
|  | **Tussock** | **13.65** | **0.003** |

Note:
The significance of the variables was determined using a likelihood ratio test, compared to the chi-square distribution. Bold letters indicate categories which differed among sika deer establishment years ($P < 0.05$). Sample size is 440 plots.

(Table 1). In perennials, the coverage at the 2011-year sites was higher than that in the 1978-year sites (Fig. S1A). For vines, cover was lower in 1978-year sites than that in the 2014-year sites. For shrubs, cover was higher in the 2014-year sites than that in the other sites, and was higher in the 2003-year sites than that in the 1978-year sites. In the category of plant architecture, it was higher in the 2011-year sites than that in the 2003-year sites for prostrate plants (Fig. S1B). For tussock, cover was lower in the 1978-year sites than that in the other sites.

Plant community composition differed among the establishment years ($R^2 = 0.21$, $P = 0.001$, Fig. 3). We did not detect spatial correlation, as none of the vectors (MEMs)
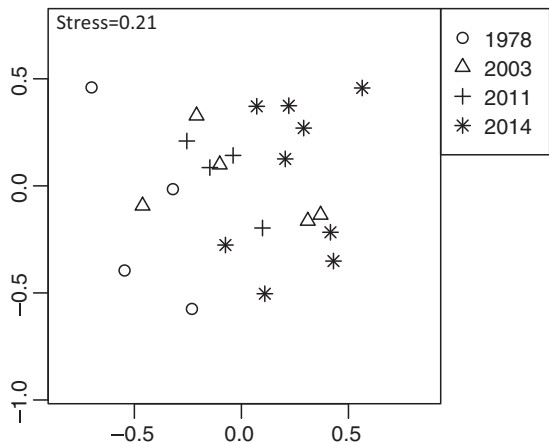

**Figure 3 Differences in plant community composition including 273 species from 22 sites with four establishment years of deer using non-metric multidimensional scaling (NMDS).** Different symbols indicate sites with different establishment year of sika deer. Analyses were based on Bray–Curtis dissimilarity in the coverage of each plant species. Stress is a statistic of goodness of fit, and it is a function of non-linear monotone transformation of observed dissimilarities and ordination distances (*Okansen, 2015*), where stress trends towards zero when the rank orders reach perfect agreement.

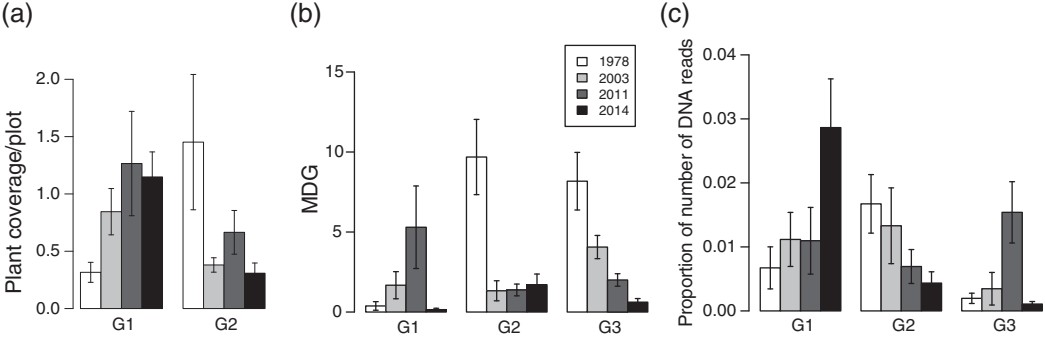

**Figure 4 Plant vulnerability groups defined with K-means analyses.** (A) plant coverage, (B) MDG (mean damage grade) of browsing damage, and (C) DNA barcoding of feces. Plants categorized in the same group show similar patterns of variation in each category among deer establishment years. Values indicate mean ± SE of each group among sika deer establishment years.

were statistically significant. The 1978-year sites were plotted separately from other sites. The plant coverage of the four establishment years combined differed among life forms ($R^2 = 0.13$, $P = 0.012$), and plant architecture ($R^2 = 0.24$, $P = 0.029$). The clustering analysis classified species into the following two groups: (1) plants whose coverage decreased in sites with longer establishment years, and (2) plants whose coverage did not decrease in sites with longer establishment years but was highest in the 1978-year sites (Table S4, Fig. 4A).

### *Browsing damage of plants*

Browsing damage was detected on 174 plant taxa in total across all sites. Plant coverage had a weak positive effect on MDG (MDG = exp (0.0027*coverage + 1.109), $\chi^2 = 4.15$,

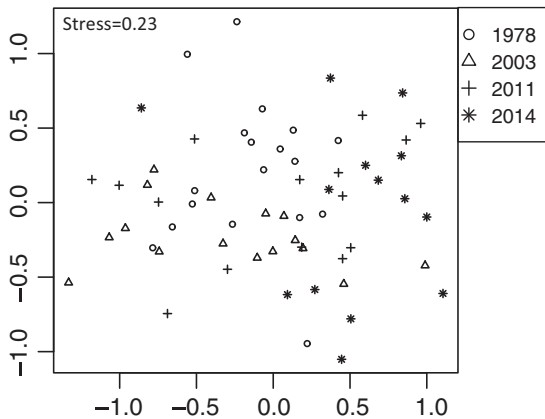

**Figure 5 Differences in sika deer diet determined by fecal analysis including 63 feces sampled in 16 sites using NMDS.** Analyses were based on the Bray–Curtis dissimilarity in the read rates of individual samples. See Fig. 3 for explanation of stress values.

$P = 0.04$, $N = 724$). The MDG of the four combined establishment years did not differ either between life forms ($R^2 = 0.18$, $P = 0.06$), or plant architecture ($R^2 = 0.18$, $P = 0.06$). The clustering analysis classified 34 species into three groups, G1–G3 (Table S4, Fig. 4B), and we categorized 16 species that were only absent in the 1978-year sites as G4 (Table S4). G2 consisted of plants whose MDG values were constantly high at the 2014-, 2011-, and 2003-year sites, and was especially high at the 1978-year sites than that at the other sites (Fig. 4B). G3 comprised plants in which the MDG value gradually increased as the establishment years increased. G1 consisted of four species that were not classified into the former two groups and showed higher values of MDG in the 2011- and 2003-year sites than those at other sites. In G4, shrubs excluding *Corylus sieboldiana* had high MDG values in all three establishment years, and perennials were browsed in the 2003- and 2011-year sites but not in the 2014-year sites, and all belonged to an erect growth form.

### DNA barcoding of feces

Sequencing of 63 fecal samples yielded 424,114 reads, corresponding to an average of 74.7 bp. From the results of DNA barcoding with the P6 loop database, 98 plant taxa (49 families) were detected from 401,206 reads (95% of the total sequences) (Table S2). Most of the reads that were not assigned to a specific plant taxon in the P6 loop database (22,908 reads) were short or had a low frequency. The sequence file of the fecal sequences was deposited at the International Nucleotide Sequence Database Collaboration through the DNA Data Bank of Japan (Accession no. DRA011125).

Out of 98 plant taxa detected from the 63 feces samples, 73 plant taxa were detected throughout the four establishment years. In 1978, 2003, 2011, and 2014 year sites, 44, 36, 35, and 36 plant taxa were detected, respectively. We did not detect spatial correlation, as none of the vectors (MEMs) were statistically significant. Diet composition (proportion of reads of plant taxa) differed among the establishment years ($R^2 = 0.13$, $P < 0.001$, Fig. 5). This was significant even when feces sampled in November were excluded from the

**Table 2 Results of the GLMM explaining the effect of deer establishment year on proportion of number of DNA reads in deer feces of each plant category.**

| | | Establishment year (d.f. = 1) | |
|---|---|---|---|
| | | $\chi^2$ | $P$ |
| Life form | Annual/biannual | 2.01 | 0.08 |
| | Perennial | 1.40 | 0.23 |
| | Vine | 0.00 | 0.99 |
| | **Tree** | **6.59** | **0.01** |
| | **Shrub** | **4.86** | **0.03** |
| Plant architecture | Prostrate | 0.48 | 0.49 |
| | Erect | 0.31 | 0.58 |
| | Rosset | 0.003 | 0.95 |
| | Rosset (seasonal) | 0.32 | 0.56 |
| | Tussock | 2.34 | 0.13 |

Note:
The significance of the variables was determined using a likelihood ratio test, compared to the chi-square distribution. Bold letters indicate categories which differed among sika deer establishment years ($P < 0.05$). Sample size is 63 feces samples.

analysis ($R^2 = 0.13$, $P = 0.001$). However, when the site was considered as a block effect, the effect of establishment years was not significant ($R^2 = 0.13$, $P = 0.99$), indicating that the composition of the feces varied among sites. In particular, the 2014-year sites were plotted differently from other establishment year sites (Fig. 5). In the 2014-year sites 26 species were absent or in remarkably low frequency (Table S5). These plants include small plants shorter than 20 cm or rosette plants such as *Hydrocotyle* spp., *Stellaria uliginosa*, *Ajuga* spp., *Oxalis corniculata*, *Lysimachia japonica*, and *Viola vaginata*. Additionally, the number of reads in some shrubs was exceptionally higher in the 2014-year sites than that at other sites such as *Stachyurus paraecox*, *Aucuba japonica*, *Rosa multiflora*, *Kerria japonica*, and *Euonymus alatus* var. *altus*. Within each category of life forms and plant architectures, there were no significant differences among the establishment years except for trees and shrubs (Table 2, Fig. S2). The number of reads of the trees gradually increased with a greater number of years since establishment, and it was significantly higher in the 1978-year sites than that in the 2014-year sites (Fig. S2A). Conversely, the number of reads of shrubs gradually decreased as the establishment years extended, and it was significantly lower in the 1978-year sites than that in the 2014-year sites.

The proportion of reads of plant taxa of the four establishment years combined did not differ either among lifeforms ($R^2 = 0.04$, $P = 0.84$) or among plant architectures ($R^2 = 0.18$, $P = 0.06$). The clustering analysis classified species into three groups G1–G3 (Table S5, Fig. 4C): G1 included plants in which the proportion of reads in the 1978-year sites was higher than that in the other year sites, G2 included plants in which the proportion of reads became gradually higher as establishment years became longer, and G3 was

species that was not classified into the former two groups and showed a higher proportion of reads in the 2011 year-sites (Fig. 4C).

### Assessment of vulnerability of plant species

We integrated the results of the three cluster analyses described above, and 98 plants that were included in either MDG analyses or DNA barcoding in feces were categorized into six groups (Table 3, group A–E). Please see Appendix 3 for detailed description of categorization of the groups.

## DISCUSSION

The results of the browsing damage and diet analysis using DNA barcoding indicated that deer selection may vary among deer establishment years. The categorization based on the three cluster analyses showed that 16 species (group A) were browsed at a greater rate from the initial stage of the deer establishment and showed lower coverage in sites with more than 10 years since deer establishment. The 16 plant species belonged to various life forms and plant architectures indicating that these indices might not always be effective in predicting plant vulnerability to browsing. Moreover, 11 of the 16 species were detected both in the browsing damage survey and DNA barcoding of feces, suggesting that deer exhibited a high selection for these species. Conversely, it was revealed that 14 species (group C) were not as highly browsed from the initial stage of deer establishment as group A, but showed lower coverage in sites with more than 10 years since deer establishment. On the other hand, 11 species (groups E and F) were selected but displayed negligible browsing effects as their coverage did not differ among deer establishment years. These results are consistent with those reported by *Akashi, Unno & Terazawa (2015)*, who stated that although the level of browse was the same, recovery differed among different species in juvenile trees. This may be because tolerance to browsing differs among species. Noteworthy, these categorizations of plant vulnerability based on degree of damage and coverage in the summer among deer establishment years might differ in other seasons as studies have reported seasonal changes in deer diet both in medium and severely damaged forests (*Takatsuki, 1986*; *Nakahama et al., 2020*). *Takatsuki (1986)* studied the deer diet at Mt. Goyo in Japan, which is located in the region of our study, and showed that *Sasa* spp. was the most dominant plant species on the deer diet and its proportion increased, especially in the winter and early spring. Further study of seasonal changes in deer diet is necessary to better understand vulnerability to browsing among plant species.

Vegetation coverage and number of species was lower only in the 1978-year sites. These results are consistent with the reports of previous studies showing declines in species richness in other forest in Japan after more than 20 years of browsing (*Fujiki & Takayanagi, 2008*; *Takatsuki, 2009a*; *Tamura et al., 2011*). The coverage of perennials and vines was only low in the 1978-year sites compared to other establishment year sites. However, the coverage of shrubs was lower in 2011-year sites than in 2014-year sites, suggesting that shrubs are more vulnerable to browsing and cover may decline in the early stage of deer establishment. In the plant architecture category, while coverage did not differ among establishment years for erect, rosette plants, the coverage of tussock plants

**Table 3  98 plant species categorized into six groups differing in vulnerability to sika deer browsing.**

| Category | Family | Species | Life form | Plant architecture |
|---|---|---|---|---|
| A | Actinidiaceae | **_Actinidia arguta_** | vine | |
| A | Asteraceae | **_Ainsliaea apiculata_** | perennial | errect |
| A | Asteraceae | **_Eupatorium makinoi_** | perennial | errect |
| A | Asteraceae | _Petasites japonicus_ | perennial | rosset (seasonal) |
| A | Fabaceae | _Hylodesmum podocarpum_ | perennial | errect |
| A | Fabaceae | _Wisteria floribunda_ | vine | |
| A | Fagaceae | _Quercus crispula_ | tree | |
| A | Fagaceae | _Quercus serrata_ | tree | |
| A | Lamiaceae | **_Callicarpa japonica_** | shrub | |
| A | Lauraceae | **_Lindera umbellata var. membranacea_** | shrub | |
| A | Rosaceae | **_Kerria japonica_** | shrub | |
| A | Rosaceae | **_Rubus crataegifolius_** | shrub | |
| A | Sapindaceae | _Acer amoenum_ | tree | |
| A | Sapindaceae | **_Acer pictum_** | tree | |
| A | Sapindaceae | _Acer sieboldianum_ | tree | |
| A | Staphyleaceae | **_Staphylea bumalda_** | shrub | |
| B | Adoxaceae | _Viburnum furcatum_ | shrub | |
| B | Anacardiaceae | _Toxicodendron trichocarpum_ | shrub | |
| B | Asteraceae | **_Rudbeckia laciniata_** | perennial | rosset (seasonal) |
| B | Betulaceae | _Corylus sieboldiana_ | shrub | |
| B | Celastraceae | _Celastrus orbiculatus_ | shrub | |
| B | Colchicaceae | _Disporum smilacinum_ | perennial | errect |
| B | Ericaceae | _Rhododendron kaempferi_ | shrub | |
| B | Garryaceae | _Aucuba japonica var. borealis_ | shrub | |
| B | Hydrangeaceae | _Heteromalla paniculata_ | shrub | |
| B | Lythraceae | _Lythrum anceps_ | perennial | |
| B | Moraceae | _Morus australis_ | shrub | |
| B | Pinaceae | _Larix kaempferi_ | tree | |
| B | Plantaginaceae | _Plantago asiatica_ | perennial | rosset |
| B | Polygonaceae | _Persicaria thunbergii_ | annual/biennial | errect |
| B | Rhamnaceae | _Berchemia racemosa_ | vine | |
| B | Rosaceae | _Neillia incisa_ | shrub | |
| B | Rosaceae | **_Padus grayana_** | tree | |
| B | Rosaceae | _Rosa multiflora_ | shrub | |
| B | Saxifragaceae | _Rodgersia podophylla_ | perennial | errect |
| B | Stachyuraceae | _Stachyurus praecox_ | shrub | |
| B | Urticaceae | **_Boehmeria silvestrii_** | perennial | prostrate |
| C | Adoxaceae | **_Viburnum dilatatum_** | shrub | |
| C | Araliaceae | _Aralia cordata_ | perennial | errect |
| C | Araliaceae | _Kalopanax septemlobus_ | tree | |
| C | Asteraceae | **_Aster microcephalus_** | perennial | rosset (seasonal) |

| Category | Family | Species | Life form | Plant architecture |
|---|---|---|---|---|
| C | Asteraceae | *Parasenecio farfarifolius var. bulbifer* | perennial | errect |
| C | Hydrangeaceae | **Hortensia cuspidata** | shrub | |
| C | Hydrangeaceae | *Schizophragma hydrangeoides* | vine | |
| C | Lamiaceae | *Clinopodium* sp. | perennial | prostrate |
| C | Oleaceae | **Fraxinus sieboldiana** | tree | |
| C | Poaceae | *Sasa* sp. | perennial | tussock |
| C | Rosaceae | **Rubus palmatus** | shrub | |
| C | Saxifragaceae | **Astilbe thunbergii** | perennial | errect |
| C | Ulmaceae | *Zelkova serrata* | tree | |
| C | Urticaceae | *Elatostema involucratum* | perennial | prostrate |
| D | Adoxaceae | *Viburnum opulus var. sargentii* | shrub | |
| D | Anacardiaceae | *Rhus javanica* | shrub | |
| D | Aquifoliaceae | *Ilex macropoda* | tree | |
| D | Asteraceae | *Artemisia indica* | perennial | errect |
| D | Asteraceae | **Aster savatieri** | perennial | errect |
| D | Caprifoliaceae | *Abelia spathulata* | shrub | |
| D | Caprifoliaceae | *Weigela hortensis* | shrub | |
| D | Caryophyllaceae | *Stellaria uliginosa var. undulata* | perennial | |
| D | Celastraceae | *Celastrus orbiculatus* | vine | |
| D | Celastraceae | *Euonymus alatus var. altus f. striatus* | shrub | |
| D | Clethraceae | *Clethra barbinervis* | tree | |
| D | Commelinaceae | *Commelina communis* | annual/biennial | prostrate |
| D | Cornaceae | *Cornus controversa* | tree | |
| D | Cupressaceae | *Chamaecyparis obtusa* | tree | |
| D | Fabaceae | *Amphicarpaea edgeworthii* | vine | |
| D | Fabaceae | *Lespedeza bicolor* | shrub | |
| D | Fagaceae | *Castanea crenata* | tree | |
| D | Juglandaceae | *Juglans mandshurica* | tree | |
| D | Lamiaceae | *Ajuga* sp. | perennial | |
| D | Oxalidaceae | *Oxalis corniculata* | perennial | prostrate |
| D | Polygonaceae | *Persicaria muricata* | perennial | |
| D | Primulaceae | **Lysimachia fortunei** | perennial | errect |
| D | Primulaceae | *Lysimachia japonica* | perennial | |
| D | Rhamnaceae | *Hovenia trichocarpa* | tree | |
| D | Rosaceae | *Aria alnifolia* | tree | |
| D | Rosaceae | *Cerasus* sp. | tree | |
| D | Rosaceae | *Potentilla centigrana* | perennial | |
| D | Rosaceae | *Rubus mesogaeus* | shrub | |
| D | Rosaceae | *Sorbus commixta* | tree | |
| D | Sapindaceae | *Aesculus turbinata* | tree | |
| D | Schisandraceae | *Schisandra repanda* | vine | |

(Continued)

| Category | Family | *Species* | Life form | Plant architecture |
|---|---|---|---|---|
| D | Ulmaceae | *Ulmus davidiana* | tree | |
| D | Ulmaceae | *Ulmus laciniata* | tree | |
| D | Violaceae | *Viola vaginata* | perennial | rosset |
| D | Vitaceae | *Ampelopsis glandulosa* | vine | |
| D | Vitaceae | *Vitis flexuosa* | vine | |
| E | Sapindaceae | ***Acer rufinerve*** | tree | |
| E | Liliaceae | *Tricyrtis affinis* | perennial | errect |
| F | Apiaceae | *Torilis japonica* | annual/biennial | errect |
| F | Asteraceae | *Solidago virgaurea* | perennial | rosset (seasonal) |
| F | Betulaceae | ***Carpinus cordata*** | tree | |
| F | Betulaceae | *Carpinus laxiflora* | tree | |
| F | Polygonaceae | *Persicaria debilis* | annual/biennial | errect |
| F | Polygonaceae | ***Persicaria filiformis*** | perennial | errect |
| F | Urticaceae | *Laportea bulbifera* | perennial | prostrate |
| F | Urticaceae | ***Laportea cuspidata*** | perennial | prostrate |
| | Chloranthaceae | *Chloranthus serratus* | perennial | errect |

**Note:**
Plant species in bold letters indicate species that were detected in both browsing damage and DNA barcoding of feces. Plant species with underline letters indicate plants that were present except in 1978 year site. A: plant species browsed in initial establishment years and showed lower coverage in sites with more than 10 years since deer establishment; B: plant species browsed in initial establishment year but the coverage difference among deer establishment was unknown; C: plant species which suffered low browsing level in initial establishment years (less than ten years) but high browsing in the latter years, and showed lower coverage in sites with more than 10 years since deer establishment; D: plant species which suffered browsing in the latter years but their coverage differences among deer establishment years was unknown.

was considerably lower in the 1978-year sites but prostrate plants showed lower coverage in the 2003-year sites. Previous studies reported that long lived perennials and- erect plants are likely to be more affected by browsing than annuals, and rosette plants (*Diaz et al., 2007*; *Tamura et al., 2011*), and dwarf shrubs are less abundant toward the high end of the herbivory gradient than juvenile trees (*Hegland & Rydgren, 2016*). Our results partly support these reports; however, the large variation we observed suggests that the response to deer establishment year differs at the plant species level within each category.

The weak positive effect of coverage on the MDG of each species suggests that deer are likely to feed on the more dominant plants as their foraging availability is known to depend on plant density (*Boulanger et al., 2009*), and plant apparency (*Furedi, 2004*). Although reduced apparency can delay ungulates detecting a plant, a study has shown that wallabies tend to browse more on lower biomass plants due to visual cue of younger plants or new growth in a plant (*Stutz et al., 2017*). Thus, plant apparency may not always correspond to the level of herbivory. Unlike the vegetation coverage, plant life form and plant architecture did not explain the pattern of deer food selection among establishment years, as shown by the large variance of MDG within each establishment year. The nutritional value and chemical content of plants determine the forage selection of deer (*Champagne et al., 2020*). Thus, plant resistance based on phytochemicals, which may vary among plant life forms and plant architectures, is likely to be more important than plant apparency for food selection by deer. However, many of the shrubs were

classified as G2 and G4 (browsed in 2003, 2011, and 2014 but absent in 1978-year sites). In addition, the perennials classified as G2 and G4 were erect in growth form. Diet analysis using DNA barcoding also showed that shrub species were detected at a much higher frequency in feces from the 2014-year sites than the other sites. These results suggest that shrubs and erect herbs may be more heavily browsed than other plants, and some may disappear in forests where browsing occurred for more than 20 years. Conversely, the average proportion of rosette plant taxa detected in the DNA barcoding was 2.27 in the 1978-year sites, while it was 0.93, 0.53, 0.92 in 2003-, 2011-, and 2014-year sites, respectively (Table S5). This may reflect the deer forage on rosette plants more in forests where browsing occurred for more than 20 years. This may be explained by the nearby presence of other more attractive species in the initial deer established sites where the rich plant community remained, but not in the severely browsed sites (*Boulanger et al., 2009*).

The diet analysis using DNA barcoding provided high-resolution identification of food plants and detected 28 plant taxa that were included in the analysis of browsing damage. Thus, DNA barcoding is a useful method for detecting a wide array of plant species within the feeding range of deer. However, 21 plant taxa included in the browsing damage analysis were not detected by the DNA barcoding. This discrepancy may be due to insufficient sampling of feces or due to issues related to low PCR amplification in some species, which is a future challenge for the use of this method (*Nichols, Åkesson & Kjellander, 2016*).

The species composition of the understory vegetation varied with deer establishment year. These results indicated that although the sum of cover and number of species did not change until decades of browsing had occurred, the species composition can change across a much shorter time scale. However, we acknowledge the limitation of our study as we did not account for the difference in the plant community among sites before deer establishment. Because there is a longitudinal cline in deer establishment years, we could not separate the effect of deer establishment year from abiotic factors such as snow depth (greater snow depth in Japan seaside than in Pacific seaside) on the difference in vegetation among sites (*Takatsuki, 2009b*). The DNA barcoding of feces indicated that the diet of deer largely differed among sites and we could not exclude the possibility that deer diet is more affected by site specific factors. However, it is noteworthy that the plant community differed even when we limited our data analysis to Pacific seaside sites ($R^2 = 0.12$, $P = 0.004$), indicating that the difference in the vegetation between Pacific seaside and Japan seaside resulting from snow depth difference is less likely to explain the plant community differences observed in our sites. In addition, we cannot deny the possibility that the deer food selection had been influenced by the initial composition of the plant community which may had led to different trajectories in deer impacts on the ecosystem as deer food choices are known to be influenced by the plant species composition (*Champagne et al., 2018*). Considering data from the plant community in sites without deer invasion and/or monitoring the temporal changes of the plant community in the recent established sites are essential in future studies.

Previous studies have suggested the use of indicator species, which is an approach that monitors plant size or browsing damage of a specific species selected by ungulates, to predict ungulate abundance (*Augustine & DeCalesta, 2003*; *Mysterud et al., 2010*; *Akashi, Unno & Terazawa, 2015*; *Iijima & Nagaike, 2015*; *Inatomi, Uno & Iijima, 2017*). *Waller, Johnson & Witt (2017)* have also shown that twig ages provide a direct indicator of browsing on regenerating trees with lower sampling variance, greater sensitivity, and are reliable indicator of deer impacts and habitat conditions. These studies have shown that these indicators may be effective in gaining information not only when the ungulate population is increasing, but also when it decreases after management. The categorization of plant species vulnerability to deer browsing identified by multiple aspects of the present study provides insight into the selection of indicator species. Selecting appropriate indicator species would allow evaluation of browsing impact before and after management. Our results strongly emphasize the need for more studies of temporal changes in plant abundance and food selection by deer at the local scale, which would allow more effective conservation of forest vegetation. This would also allow better evaluation of the validity of using a chronosequence of deer establishment years to predict changes in plant community composition caused by deer browsing.

## CONCLUSIONS

By combining multiple approaches to investigate deer diet and vegetation, we categorized 98 plant taxa into six groups that reflect vulnerability to deer browsing. In future studies, a clear separation between the effects of deer browsing and other environmental factors is necessary for a precise prediction of forest plant community composition. The differing responses to browsing among plant species inferred from this study could be a first step in predicting the short- and long-term responses of forest plant communities to deer browsing.

## ACKNOWLEDGEMENTS

We are grateful to T. Aikawa for help on the discrimination of deer feces using LAMP method and field survey on searching for deer feces. We thank staff at the district forest office in Tohoku region in Japan for guiding the field sites. We thank T. Furuta for advice on DNA barcoding analysis.

### Funding

This work was supported by the research grant of Akita Prefectural University.
The funders had no role in study design, data collection and analysis, decision to publish, or preparation of the manuscript.

### Grant Disclosures

The following grant information was disclosed by the authors:
Akita Prefectural University.

## Competing Interests

The authors declare that they have no competing interests.

## Author Contributions

- Yuzu Sakata conceived and designed the experiments, performed the experiments, analyzed the data, prepared figures and/or tables, authored or reviewed drafts of the paper, and approved the final draft.
- Nami Shirahama performed the experiments, analyzed the data, prepared figures and/or tables, authored or reviewed drafts of the paper, and approved the final draft.
- Ayaka Uechi performed the experiments, authored or reviewed drafts of the paper, and approved the final draft.
- Kunihiro Okano performed the experiments, authored or reviewed drafts of the paper, and approved the final draft.

## Field Study Permissions

The following information was supplied relating to field study approvals (*i.e.*, approving body and any reference numbers):

Field experiments were approved by Tohoku Regional Forest Office (approval number: 29-392, 29-487, 28-446).

## DNA Deposition

The following information was supplied regarding the deposition of DNA sequences:

The sequence of the trnL P6 loop of the plant Gaza and the sequence file of the fecal sequences are available at the International Nucleotide Sequence Database Collaboration through the DNA Data Bank of Japan (Accession numbers LC586452–LC586485, LC586838, and accession numbers DRA011125).

## Data Availability

The raw data are available in Supplemental Files 1–3. The raw data are showing plant species detected in the vegetation survey, feeding traces and DNA barcoding of feces.

## Supplemental Information

Supplemental information for this article can be found online at http://dx.doi.org/10.7717/peerj.12165#supplemental-information.

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
