# Peer review of "Variability in deer diet and plant vulnerability to browsing among forests with different establishment years of sika deer"

_PeerJ, doi:10.7717/peerj.12165_

## Round 0.1 · original submission · Major Revisions

Both reviewers saw merit in the manuscript and concluded that it could be suitable for publication following a major revision. I concur with their very thorough assessments. The authors should pay close attention to, and carefully address, the comments of both reviewers. Both reviewers raised issues about statistical analyses, and the conclusions based upon these analyses. Suggestions are also provided to improve clarity and consistency in the use of terminology. Because both reviewers are recognized experts in ungulate herbivory, their comments are highly relevant, and addressing them will greatly improve the manuscript.

Reviewer 1 ·

Basic reporting

This manuscript is clear, although several word choices are problematic (see general comments). The introduction presents the concept clearly, although some clarifications are required (see detailed comments in the attached PDF). The figures and tables labels provide insufficient information and need to be expanded to be more self-standing. Here’s some of the things I would need to see added either to the figures or to their captions:
• Figure 1: What are the numbers on the map? The sites? The distance scales only show two subdivisions, at an ineligible font size, making them of little help.
• Figure 2: What statistical test was used to produce the different grouping? What alpha level? Which deer species? How many plots in each category of establishment years?
• Figure 3: What does an NMDS stand for? How many species considered? How many sites? What does the ‘stressed = 0.21’ at the top of the graph means? What are the labels?
• Figure 4: What are those groups? What do they represent? What does MDF stand for? Axis could also be adjusted for less white space at the top
• Figure 5: How many deer/samples? Which deer species? Again, what is this ‘stress = 0.23’ at the top of the graph?
• Tables : More information on the statistical analyses in the captions are required, and sample sizes and degrees of freedom are needed
Yes, adding this information would make repetitions from the method section, but making the figures self-standing will improve the manuscript, and especially the clarity and ease of reading.

Experimental design

This manuscript report primary research in the scope of PeerJ. The subject is interesting and relevant : the authors use a chronosequence of sika deer establishment eras to evaluate their effect on plant community cover and diversity. They also use feeding traces and deer feces to evaluate how use of plant species varies among regions with different establishment eras. The methods seem appropriate, and although I’m not aware of the standards regarding metabarcoding analyses, this seems to be well explained and appropriate. The appendices, however, could be improved for clarity (e.g. ‘we sampled the leaves of those plants’, without precision about what are those plants.). The authors should remember that it is an independent document, and asking the reader to go back and forth between two documents to understand it too much. I’m also surprised by how few references are cited in the appendix.
I have one main issue with the experimental design, in regards to the evaluation of feeding trace. Feeding traces (both browsing and grazing) were evaluated in one season, which could influence the detectability of traces. For example, if the browsing occurred in the previous winter, browsing trace on woody plants will be much harder to evaluate in the fall. I’m also wondering how the four classes of damage used can be applied to grazing on forbs and grass. I’m suspecting that this method of damage evaluation presents a strong bias among plant life forms, suggesting that the following analyses comparing life forms are biased.

Validity of the findings

This is my main concern regarding this manuscript: I do not think the interpretation and conclusions are supported by the results. This dataset is a chronosequence based on deer establishment eras, and while this type of study is useful to trace back the effect of a perturbation, the authors go too far when they extrapolate changes in deer selectivity (wrongly called palatability) among these eras.
First, those eras are very variable in size and relatively uncertain, which is obscured by the phrasing of ‘establishment year’. The second class (‘2003’) is especially large, which represent deer establishment between 1979 and 2003. The very recent (2012-2014) and very old (1978) make for a more certain comparison. Second, we have no information about the vegetation cover, composition or state before establishment. This is a very strong assumption to make that the only thing affecting vegetation changes between 1978 and 2014 is deer establishment. The authors briefly acknowledge this in the discussion, but don't really adjust their conclusions. One element of deer selectivity that is not discussed in this manuscript is how the species composition, not only the diversity or richness, can influence food choices (for example: Champagne et al. 2018). Differences in initial composition or vegetation state could have made for different trajectories in deer impacts on the ecosystem. Finally, the feces were collected once, in a seasonal ecosystem. This strongly limit the inferences that can be made on deer diet.
This doesn’t undermine the good research that was done here, but I do think the authors need to revise their conclusions. The leap to changes in forage selection is too big and the authors should be more careful in their interpretation. They also cannot infer to the tolerance of these species. Tolerance is the regrowth process following damage, and the authors rather evaluate the persistence of species in a chronosequence. The last paragraph of the discussion regarding the use of indicator species could be an interesting outcome of this study, but the link to the actual results is thin and could be clarified.
Other smaller points:
• The authors should verify if their use of Mantel test is appropriate or not based on Legendre et al. (2015).
• I tried running the code provided by the authors, and it didn’t work. I suspect the mistake is only minor: the plant_site.csv data provided has 276 variables and the first coding line selects columns from 7 to 292, which is impossible. I also don’t know what the columns name stand for.
• The sentence in lines 373-375 simplifies too much sika deer forage selection. Although plant density does affect forage selection, the support for plant apparency in mammalian herbivores is dubious… while reduced apparency can increase the time to browsing, the most profitable resources are generally consumed in the end (Stutz et al. 2015, Stutz et al. 2017). Also, there’s no surprise in not seeing a clear pattern for one plant category or another, as the nutritional value and chemical content of individual species are very determinant in forage selection (Examples of Cervus nippon selection based on chemical content can be found in Champagne et al. 2020).

Additional comments

As pointed out in a previous section, I disagree with the use of certain terms. First, palatability is a very vague concept and the authors should define it or rather use a more appropriate term. Are they referring to the chemical composition, the chemical value or to the use or selection by sika deer? I suggest reading the classic Johnson (1980) for an appropriate use of the latter. It’s especially important for this manuscript to use the proper terms, because I doubt plant palatability actually changed. It’s the behaviour of the animals that potentially did, and what they consider to be an acceptable resource, depending on what is available.
The authors also frequently refer to ‘vegetation’ but it’s not what always clear what element of vegetation community is referred to. For example, in lines 17-18: ‘the differences in plant palatability and vegetation among different stages’. Are we talking about changes in composition, diversity, growth rates, abundance (the ‘coverage’)? The ‘number of plant species’ is actually the richness.
References
Champagne, E., A. Dumont, J.-P. Tremblay, and S. D. Côté. 2018. Forage diversity, type and abundance influence winter resource selection by white-tailed deer. Journal of Vegetation Science 29:619-628.
Champagne, E., A. A. Royo, J.-P. Tremblay, and P. Raymond. 2020. Phytochemicals involved in plant resistance to leporids and cervids: a systematic review. Journal of Chemical Ecology 46:84-98.
Johnson, D. H. 1980. The comparison of usage and availability measurements for evaluating resource preference. Ecology 61:65-71.
Legendre, P., M. J. Fortin, and D. Borcard. 2015. Should the Mantel test be used in spatial analysis? Methods in Ecology and Evolution 6:1239-1247.
Stutz, R. S., P. B. Banks, N. Dexter, and C. McArthur. 2015. Herbivore search behaviour drives associational plant refuge. Acta Oecologica 67:1-7.
Stutz, R. S., B. M. Croak, N. Proschogo, P. B. Banks, and C. McArthur. 2017. Olfactory and visual plant cues as drivers of selective herbivory. Oikos 126:259-268.

Annotated reviews are not available for download in order to protect the identity of reviewers who chose to remain anonymous.

Reviewer 2 ·

Basic reporting

Authors report on a field study aimed at understanding deer diet at sites that differ in time since deer establishment. At each site, researchers documented understory plant communities and feeding traces on target plant species. In addition, they use DNA barcoding techniques to identify plant species consumed by deer and present in their feces. Results indicate a significant shift in plant communities and deer preferences according to time since deer establishment. The results may expand our understanding of deer impacts and inform management. Before publication data analyses should be revised. Please see my comments below.

1. Please develop hypothesis and predictions for the study and include them at the end of the introduction
2. I suggest authors review newer literature. For example, most papers cited on lines 33-39 are more than 20 years old. Newer literature has summarized current impacts of ungulate introduction or increased abundance. Similarly, on line 49, although there are technical difficulties with density models for deer, there are newer applications and developments.

Experimental design

1. Lack of independence in statistical analyses. How did authors account for the lack of independence among quadrats within the same site? It is not clear, if they entered a single value per site or all replicates per site. Replicates within the same site are not independent and treating them as independent would violate model assumptions. Please clarify and/or re-run analyses to avoid violation of independence assumption. Same consideration applies to species richness, diversity indexes and PERMANOVA analyses.
2. Several tests were performed, but the purposes of each is not clear, nor why certain PERMANOVA were conducted twice (both with the same response variable). The reasoning for each analysis is not clear. Please revise writing of this section

Validity of the findings

1. Given the ambiguity in the analysis, it not possible to evaluate the validity of findings. Findings might change once analyses are re-run to account for lack of independence among samples or the text is clarified.
2. As described, the methodology of the feeding trace component cannot determine if feeding was due to deer browsing or other herbivores. Also, it is not clear how local abundance of each species was determined. I suggest authors clarify the writing and describe how they excluded herbivory by other mammals (for example, rodents). If consumption by other herbivores was not recorded or separated during data collection, this section needs to be revised.

Additional comments

3. L56-58 Please revise language. Meaning is not clear
4. L103: should be “all area in Japan is divided”, not “are divided”. There is another example with verb agreement on line 164 (were applied not was applied). Please revise the whole document
5. L107: can authors explain why did they not select sites within category 5. Wouldn’t absence of deer be considered a control or reference for vegetation communities?
6. L114-115: was each site sampled twice or just once? If only once, were sites sampled on different years, and how was this accounted for? Where all sites 500x500m?
7. L120: while ferns are not regularly eaten by deer, their abundance may increase under high deer abundance. In North America high fern abundance is associated with increased deer activity. Authors should discuss this aspect and review literature regarding fern abundance
8. L126: By convention, numbers at the beginning of a sentence should be spelled (ten not 10)
9. L129: were individuals selected at random? Was the whole area (500 x 500 m) sampled? How far apart were individuals? Please specify. Also, specify how you estimated average plant height. Wouldn’t height be affected by deer activity? And consider reporting the average plant height at each site in the supplementary
10. L124 and related text: clarify that feeding damage is not deer damage. The feeding damage may have been produced by any herbivore.
11. L134-135. How was the total number of ramets at each site estimated? Was this estimated for the whole area site? Please provide details
12. L184-192. Please clarify which plant data you used for these analyses. Is it the understory vegetation? If it is only about the vegetation survey, why are two PERMANOVAs required?
13. L224: proportion should not be capitalized
14. L254-259. Is there one site per year-of-deer-establishment category? Or is this a typo? Also, provide evidence for claims of abundant plants, feces and trails. What is abundant? What was considered unpalatable? Report abundance of said species and provide examples or refer to supplementary table. Any analysis to support these claims?
15. Fig. 1. Please revise caption. Figure has a panel a and b, but there is no reference of that in the caption. Also, indicate that site number is correspondent to Table S1, which provides all the information about each site. Indicate in legend that categories represent years since deer establishment and identify the site at which picture was taken
16. Fig. 2. Indicate number of replicates in caption
17. Table S1: x and o symbols are confusing. I suggest replacing by 1/0, detected/not detected or yes/no. Also, include a brief description of the procedure used to detect feces
18. There is not Table S2 in supplementary file

---

## Round 0.2 · Major Revisions

While the revised manuscript is improved, the reviewer identified additional changes that are needed. The authors should address all of the reviewer comments. Also, the reviewer noted the the R code still does not run. This needs to be addressed. Finally, I have attached a version of the manuscript with editorial changes that need to be made to improve clarity. Much of the new text in the discussion and conclusions consists of run-on sentences. This text needs to be carefully read and revised for clarity.

Reviewer 1 ·

Basic reporting

The reporting of this study is generally up to the level required for publication. Some of the added/modified sentence can be a little difficult to read and should be revised.

There's also a small issue with the changes from 'palatability' to 'selectivity' which I discuss in the general comments.

While the raw data and code is shared, the code does not run (see section 3).

Experimental design

No comment.

Validity of the findings

The changes made to the manuscript add to the manuscript a good discussion of the study limitations.

However, the answer to my comment 6 has left me perplexed : 'We did not include degree of freedom because it corresponds with the sample size.'. We are talking about mixed models with a blocking structure, in which case I highly doubt that the degrees of freedom equal the sample size. Moreover, if we take the example of Table 1, this would mean there's a big issue with the determination of statistical significance, because the critical value for a chi-square distribution with a sample size of 440 is 489...and none of the presented results would be significant.

What the authors are reporting here are the results of the a posteriori test within a significant main response variable (aka life form and plant architecture). They need to report the statistics associated with the response variable (probably an F value) with the appropriate degrees of freedom (numerator and denominator).

I actually tried to run the authors code to provide them guidance on how to obtain this information. The code is not working past the line 15.

Additional comments

In general, I am satisfied with the changes made to this manuscript and to the answers provided to my previous comments.

One of my suggestion, however, was perhaps poorly described and led to some confusion. The change to 'palatability' to 'selectivity' everywhere doesn't work. Actually, the authors mean 'selection' much more often than they mean 'selectivity'. Selectivity is the degree of selection done by the deer (is it a picky eater or not), while selection is actually what was consummed. In some occurence, selectivity should be changed to selection. See example in line 96.
In others, the authors actually 'vulnerability' or 'susceptibility', that is the risk experienced by a plant species of being browsed. See example in line 58.
Finally, the first paragraph of the discussion talks about preference. Unless you have the availability of the resource, this is not preference, this is selection. To measure preference, you would need equal availability (See Johnson 1980). This is the case in lines 380-382.

Two very minor details:
L 141-143 : Perhaps explain how rabbit traces differ from deer for unaware readers.
L 305-306 'A spatial autocorrelation was not detected as none of the vectors (MEMs) were detected significant.': Bulky sentence. Rather try: We did not detect spatial correlation, as none of the vectors (MEMs) were statistically significant.

---

## Round 0.3 · Minor Revisions

This version is much improved. I have made edits to the manuscript text and figure/table captions (peerj 57993 PeerJ research manuscript AE edits2.pdf) to improve clarity. My edits are all minor and the manuscript will be acceptable for publication after they are made. Congratulations on your excellent work. I look forward to seeing it published.

---

## Round 0.4 · accepted · Accept

Congratulations on an excellent contribution to the ungulate herbivory literature.